# Bacterial and Viral Co-Infections in COVID-19 Patients: Etiology and Clinical Impact

**DOI:** 10.3390/biomedicines12102210

**Published:** 2024-09-27

**Authors:** Ivelina Trifonova, Iveta Madzharova, Neli Korsun, Viktoria Levterova, Petar Velikov, Silvya Voleva, Ivan Ivanov, Daniel Ivanov, Ralitsa Yordanova, Tatiana Tcherveniakova, Svetla Angelova, Iva Christova

**Affiliations:** 1National Laboratory “Influenza and ARD”, Department of Virology, National Centre of Infectious and Parasitic Diseases, 1504 Sofia, Bulgaria; neli_korsun@abv.bg (N.K.); vikis@abv.bg (V.L.); iva_christova@yahoo.com (I.C.); 2Infectious Disease Hospital “Prof. Ivan Kirov”, Department for Infectious Diseases, Parasitology and Tropical Medicine, Medical University of Sofia, 1431 Sofia, Bulgaria; petar.kr.velikov@gmail.com (P.V.); svoleva@abv.bg (S.V.); iri85@abv.bg (I.I.); dannieltiv@gmail.com (D.I.); yordanova_r@abv.bg (R.Y.); taniatcher@gmail.com (T.T.); 3Clinical Virology Laboratory, University Hospital “Prof. Dr. Stoyan Kirkovich”, Trakia University, 6000 Stara Zagora, Bulgaria; svetla_georgieva@abv.bg

**Keywords:** SARS-CoV-2, co-infection, respiratory viruses, *Haemophilus influenzae*, RSV

## Abstract

Background: Mixed infections can worsen disease symptoms. This study investigated the impact of mixed infections with viral and bacterial pathogens in patients positive for severe acute respiratory syndrome coronavirus 2 (SARS-CoV-2). Methods: Using the in-house multiplex PCR method, we tested 337 SARS-CoV-2 positive samples for co-infections with three bacterial and 14 other viral pathogens. Results: Between August 2021 and May 2022, 8% of 337 SARS-CoV-2-positive patients had bacterial co-infections, 5.6% had viral co-infections, and 1.4% had triple mixed infections. The most common causes of mixed infections were *Haemophilus influenzae* (5.93%) and respiratory syncytial virus (RSV) (1.18%). Children < 5 years old had more frequent co-infections than adults < 65 years old (20.8% vs. 16.4%), while adults showed a more severe clinical picture with a higher C-reactive protein (CRP) level (78.1 vs.16.2 mg/L; *p* = 0.033), a lower oxygen saturation (SpO2) (89.5 vs. 93.2%), and a longer hospital stay (8.1 vs. 3.1 days; *p* = 0.025) (mean levels). The risk of a fatal outcome was 41% in unvaccinated patients (*p* = 0.713), which increased by 2.66% with co-infection with two pathogens (*p* = 0.342) and by 26% with three pathogens (*p* = 0.005). Additionally, 50% of intensive care unit (ICU) patients had a triple infection, compared with only 1.3% in the inpatient unit (*p* = 0.0029). The risk of death and/or ICU admission was 12 times higher (*p* = 0.042) with an additional pathogen and increased by 95% (*p* = 0.003) with a third concomitant pathogen. Conclusions: Regular multiplex testing is important for prompt treatment and targeted antibiotic use.

## 1. Introduction

Globally, the coronavirus disease 2019 (COVID-19) pandemic has impacted the health and economic systems of every affected country. After the emergence of the first strain of severe acute respiratory syndrome coronavirus 2 (SARS-CoV-2), and then Alpha, Delta, and Omicron, the virus gradually acquired a strong degree of spread with increased transmissibility and infectivity [1]. Pre-Omicron variants of SARS-CoV-2 were relatively more aggressive in terms of more severe clinical presentation of COVID-19 and mortality, and this led to an overload of the hospital system [2]. This resulted in the implementation of emergency non-pharmaceutical measures, isolation, and reduced travel to contain the spread of SARS-CoV-2. These measures have impacted the spread of other respiratory bacteria and viruses. During the winter seasons of 2020/2021 and 2021/2022, reduced or no spread of influenza viruses was reported worldwide [3,4,5]. In 2018–2020, the average prevalence of respiratory viruses reported in studies ranged from 49.8% to 39%, while with the onset of the 2020–2021 COVID-19 pandemic, the prevalence was significantly reduced to 13.4% [6]. A low prevalence of respiratory viruses invariably reflects the level of evidence for respiratory co-infections. Before the pandemic, reported rates of mixed respiratory viral infections were in the order of 10% [7]. During the height of the pandemic in 2020, one study put the rate of SARS-negative patients at 4.3%, and that of positives was even lower, at 2.5% [8]. The data on the prevalence of co-infections associated with COVID-19 vary across studies: prevalences of 7% and 3% of bacterial and viral co-infections in hospitalized patients have been reported [9]. Another meta-analysis reported a 5% incidence of co-infections with SARS-CoV-2 and other respiratory viruses, noting that it was greater (approximately 9.4%) in pediatric patients than in adults (3.5%) [10]. Serious respiratory viral infections are worsened by bacterial co-infections and secondary infections, leading to higher rates of illness and death [11]. Some of the most commonly reported co-pathogens in hospitalized patients with COVID-19 are *Mycoplasma pneumoniae* (MP), *Haemophilus influenzae* (HI), and *Chlamydia pneumoniae* (ChP) [12,13,14,15].

Several studies have confirmed that a large percentage of SARS-CoV-2 infections are asymptomatic [16,17]. However, such an asymptomatic or clinically manifested SARS-CoV-2 infection in combination with an infection caused by another respiratory virus or bacteria may worsen the clinical course of the disease. A previous study by this team reported a 25% increase in the probability of a fatal outcome in SARS-CoV-2-positive patients co-infected with another respiratory virus compared with mono-infected patients [18]. In that study, we did not simultaneously examine bacterial and viral co-infections. Our study addresses this gap by evaluating the clinical burden of bacterial and viral co-infections. Previous studies have given little or no attention to the impact of accumulating co-infections compared with dual infection. This study’s uniqueness lies in evaluating the risk of concurrent infection with multiple co-pathogens compared with mixed infections with two co-pathogens.

Mixed infections can significantly weaken the body’s immune system, worsening the disease. Bacteria and viruses have a symbiotic relationship that can lead to co-infections [19]. Bacteria can promote viral infection by altering the host’s immune response, controlling surface adhesion proteins, and activating viral proteins [20]. Similarly, viruses can assist bacteria in causing secondary infections by compromising the host’s immune response, disrupting the integrity of the epithelial barrier, and expressing surface receptors and adhesion proteins [21,22,23]. Patients with pneumonia caused by COVID-19 have a clinical and radiographic presentation similar to that of other viral, fungal, and bacterial pneumonia [24]. Therefore, it is difficult for diagnosticians to distinguish the causative agents of the respiratory infection. This delays treatment and makes it difficult to choose targeted antibiotic treatment and/or antiviral therapy [25]. Understanding these mechanisms is critical to the effective management of infectious diseases. In COVID-19 patients, it is crucial to differentiate between bacterial and viral co-pathogens to provide targeted treatment while minimizing antibiotic overuse [26]. This study aimed to determine the etiology and clinical manifestations of mixed infections with bacterial and viral co-pathogens.

## 2. Materials and Methods

### 2.1. Population Survey and Sampling

Between August 2021 and May 2022, nasopharyngeal samples were collected from 337 SARS-CoV-2-positive patients. The inclusion criterion was based on a positive test for SARS-CoV-2. Emergency department patients with milder symptoms, as well as those hospitalized with severe and moderate COVID-19, were included. Severe COVID-19 outcomes were defined as defined by the United States Centers for Disease Control and Prevention (CDC) as requiring hospitalization, intensive care unit (ICU) admission, intubation or mechanical ventilation, or death (https://www.cdc.gov/covid/hcp/clinical-care/underlying-conditions.html, accessed on 26 September 2024). Patients admitted to intensive care units were also included in the study. The patient samples were collected from two different hospitals in Bulgaria: SBALPB “Prof. Ivan Kirov”, Sofia, and UMBAL “Prof. Dr. Stoyan Kirkovich”, Stara Zagora. All data on the studied patients were collected from the medical records at the hospitals. The data analyzed included symptoms, oxygen saturation, blood tests, antiviral and antibiotic use, other medications received, oxygen therapy required, the length of hospital stay, intensive care unit (ICU) admission, and fatal outcome.

Specimens for testing were collected using viral transport media from patients who visited the emergency department or were hospitalized within the first 24 h of admission and up to 7 days after the onset of respiratory symptoms. The nasopharyngeal swabs were sent to the National Laboratory “Influenza and ARD” in refrigerated conditions at 4 °C. Before shipping, the samples were stored at 4 °C for a maximum of 72 h. Upon arrival at the laboratory, the samples were processed on the same day or, if this was not possible, stored at −80 °C until analysis.

### 2.2. Detection of Respiratory Co-Pathogens

#### 2.2.1. Extraction

We used an automated extraction system with the Exi-Prep Dx Viral DNA/RNA kit from Bioneer, Daejeon, Republic of Korea, to isolate viral and bacterial DNA/RNA. We utilized 400 µL of the primary clinical material, and it took 1 h and 34 min to complete the process. The extraction system offered a choice of final volume ranging from 25 µL to 100 µL, and we selected the largest volume to avoid re-extraction.

#### 2.2.2. Detection of Bacterial Co-Pathogens

For the simultaneous detection of three bacterial pathogens—*Haemophilus influenzae* (HI), *Mycoplasma pneumoniae* (MyP), and *Chlamydia pneumoniae* (ChP)—a multiplex PCR mixture was prepared using an AgPath-ID™ One-Step RT-PCR kit (Thermo Fisher Scientific, Waltham, MA, USA) and the set of primers indicated in Appendix A [27].

A CFX96 Touch PCR system (Bio-Rad Laboratories, Inc., Hercules, CA, USA) was used. The temperature conditions for the reaction are indicated in Appendix A. The reaction lasted 1 h and 10 min.

After DNA amplification, the resulting PCR products were separated using capillary electrophoresis with a QIAxcel Advanced Automated DNA Analysis System (Qiagen, Hilden, Germany).

The resulting bands were compared with a molecular marker and positive controls for the respective bacteria acquired and confirmed using the external quality control program INSTAND (Appendix A) (the control was carried out by the National Reference Laboratory for Molecular Microbiology, annually). The expected size of the PCR products (bp) for the individual bacteria is described in Appendix A, indicating the primers used.

#### 2.2.3. Detection of Viral Co-Pathogens

A total of 6 Multiplex Real-time PCR mixes were prepared for the simultaneous detection of 8 seasonal respiratory viruses—respiratory syncytial virus (RSV), human metapneumovirus (HMPV), parainfluenza virus (PIV) types 1/2/3, rhinovirus (RV), adenovirus (AdV), and bocavirus (BoV)—and 4 endemic human coronaviruses (HCoVs) (229E, NL63, OC43, and HKU1).

The working mixes had the following combinations:(1)AdV, RSV, and PIV1;(2)BoV, RV, and PIV2;(3)HMPV and PIV3;(4)HCoV-229E and HCoV-HKU-1;(5)HCoV-NL63 and HCoV-OC43;(6)SARS-CoV-2 and influenza A and B viruses (FluSC2).

The primer and probe combinations listed in Appendix A and by previous authors [28,29] were used to prepare the PCR mixes combined with the SuperScript™ III one-step RT-PCR system with Platinum™ Taq DNA polymerase (Thermo Fisher Scientific, Waltham, MA, USA). We used FluSC2 primers and probes donated by the CDC in Atlanta, GA, USA, together with an Applied Biosystems™ TaqMan™ Multiplex Master Mix PCR System supplied by Thermo Fisher Scientific (Waltham, MA, USA) for the detection of SARS-CoV-2 and influenza viruses A and B.

The following temperature conditions were used to detect SARS-CoV-2, influenza, and other respiratory viruses:Reverse transcription: 25 °C for 2 min and then 50 °C for 15 min;Initial denaturation: 95 °C for 2 min;Amplification for 45 cycles: 95 °C for 15 s and then 55 °C for 30 s.

Samples with a cycle threshold (Ct) value of <38 were considered positive. Samples that tested positive for SARS-CoV-2 and other respiratory viruses with a Ct value below 31 were selected for NGS. The samples were re-extracted following a previously described protocol, and the Ct value was re-measured using real-time PCR.

### 2.3. Next-Generation Sequencing (NGS)

The ARTIC protocol was used to isolate the entire genome of SARS-CoV-2. NGS was performed using the Illumina MiSeq system with reagent kit v2 500 cycles (Illumina, San Diego, CA, USA). The genetic sequences were deposited in GISAID. Analysis to define SARS-CoV-2 variants was performed using the Pangolin COVID-19 Lineage Assigner v4.3 program. The quantification of the purified library pool was performed using the Qubit™ dsDNA HS Assay Kit (Thermo Fisher Scientific). The libraries underwent analysis to determine the distribution of fragment sizes using QIAxcel Advanced capillary electrophoresis (Qiagen, Hilden, Germany).

### 2.4. Definitions and Data

Pneumonia was determined using X-ray examination and the presence of pulmonary infiltrates. Acute respiratory distress was defined by the definition of Berlin [30].

The patient assessment used the following category scale (1–7) to evaluate their condition: 1—not hospitalized, with a resumption of normal activity; 2—not hospitalized, but cannot resume normal activities; 3—hospitalized, not requiring additional oxygen; 4—hospitalized, requiring additional oxygen; 5—hospitalized, requiring high-flow nasal oxygen therapy, non-invasive mechanical ventilation, or both; 6—hospitalized, requiring ECMO, invasive mechanical ventilation, or both; and 7—death.

Data from the discharge summary included demographic information, comorbidities, symptoms, lab results, and medication use. Pneumonia diagnosis was confirmed via X-ray, and acute respiratory distress syndrome was defined according to the Berlin criteria.

### 2.5. Statistical Analysis

Fisher’s exact tests or chi-square tests were used to analyze the data for categorical variables. Continuous variables were compared using the Mann–Whitney U test, mediated via the OriginPro, 2024b software and GraphPad (https://www.graphpad.com/quickcalcs/contingency1/ (accessed on 26 September 2024)). The Cox proportional hazards model was used to estimate risk, mediated via the DATAtab software (https://datatab.net/statistics-calculator/survival-analysis/cox-regression (accessed on 26 September 2024)). Probabilities were two-tailed, and *p*-values of <0.05 were considered statistically significant.

## 3. Results

### 3.1. Population Characteristics

This retrospective study spanned ten months from 2021 to 2022 and included 337 patients who tested positive for SARS-CoV-2. Of these, 277 were hospitalized (73.5%) and 100 outpatients (26.5%) with confirmed SARS-CoV-2 infection. Patients ranged in age from 3 months to 92 years (mean age: 57.8 ± 25.7), with 55% being female and 45% male. The patients were categorized into four age groups: 0–5 years (*n* = 48; 14.3%), 6–16 years (*n* = 8; 4.8%), 17–64 years (*n* = 122; 19%), and 65 years and older (*n* = 159; 61.9%).

### 3.2. Detection of Co-Infections in COVID-19 Patients

In the group of SARS-CoV-2-positive patients, 42 (12.4%) had mixed infections. Bacterial co-infections were more common than viral ones, with 27 (8%) patients having bacterial co-infections compared with 19 (5.6%) having viral co-infections (Table 1). Of the patients studied, five (1.4%) had triple infections combined with SARS-CoV-2, another respiratory virus, and a bacterial co-pathogen. *H. influenzae* was identified as the most common co-pathogen in mixed infections with SARS-CoV-2, present in 20 clinical samples (5.93%). RSV was the second most common co-pathogen, found in four cases (1.18%). Among the triple infections, the most common combination was SARS-CoV-2 + *H. influenzae* + RSV, found in three cases (0.9%).

Figure 1 shows the distribution of each proven double or triple mixed infection. PIV2, RSV, AdV, and *H. influenzae* were frequent participants in triple infections, at 100%, 42.9%, 25%, and 20%, respectively.

### 3.3. Viral Load of Respiratory Virus Co-Infections

In this study, the real-time PCR Ct value was taken as an indirect measure of viral load. We found that the viral load of SARS-CoV-2 in each case of co-infection with other respiratory viruses was significantly higher (Ct mean: 23.6 ± 3.8 vs. 36.6 ± 3; *p* < 0.001) (Figure 2).

### 3.4. Distribution of SARS-CoV-2 Variants in Individual Proven Co-Infections

SARS-CoV-2 sequencing analysis was performed in 305 (90.5%) of the patients studied. The SARS-CoV-2 Omicron (*n* = 32; 17.2%) and Delta (*n* = 2; 1.7%) variants were detected in 34 of the 42 patient samples with proven co-infection. Patients infected with the SARS-CoV-2 Omicron variant had a higher percentage of co-infections than those infected with the Delta variant (*p* = 0.0001). *H. influenzae* was also found to be the most frequent co-pathogen in patients infected with the Omicron BA.2 variant of SARS-CoV-2 (*n* = 13; 38.2%). The data in Table 2 demonstrate that a co-pathogen of bacterial origin was more frequent than that of viral origin in patients infected with Omicron (*n* = 20 (62.5%) vs. *n* = 9 (28%); *p* = 0.0081). All three triple infections found in this study were of the Omicron BA.2 subvariant. Co-infections of bacterial origin were not detected in patients with the SARS-CoV-2 Delta variant.

### 3.5. Weekly and Seasonal Distribution of Confirmed Co-Infections

The study period spanned a total of 28 weeks (August 2021–May 2022) (Figure 3), with a greater percentage of study samples collected during weeks 13–19 of 2022. The highest rates of co-infections were found during the period 13–19 weeks of 2022. During the different periods of the study, different combinations of other participants in mixed infections were observed. During weeks 34 to 42 of 2021, no mixed infections were identified. A high rate of co-infected patients with SARS-CoV-2 was demonstrated during weeks 13, 16, 18, and 19 of 2022 (March–May). The highest number of co-infections was detected in week 17 of 2022 (April 25–May 1). The same week also saw the greatest diversity in terms of proven co-infections: SARS-CoV-2 + *Haemophilus influenzae* (*n* = 5), SARS-CoV-2 + RSV (*n* = 1), SARS-CoV-2 + *Haemophilus influenzae* + AdV (*n* = 1) and SARS-CoV-2 + *Haemophilus influenzae* + PIV2 (*n* = 1).

The patients with confirmed co-infections involving SARS-CoV-2 were categorized into four age groups to analyze their age distribution: 0–5, 6–16, 17–64, and 65 years and older (Figure 4). In the group of the youngest children (0–5 years), positive for SARS-CoV-2, 6 (12.5%) cases of co-infections involving Haemophilus influenzae (*n* = 4; 8.33) %), rhinovirus (RV) (*n* = 1; 2.8%), and adenovirus (AdV) (*n* = 1; 2.8%) were proven (Figure 4). In children and adolescents aged 6–16 years, the percentage of mixed infection was the highest (25%), with the following combinations established: SARS-CoV-2 + *Haemophilus influenzae* (*n* = 1) and SARS-CoV-2 *Haemophilus influenzae* + AdV (*n* = 1) (Figure 4).

In the age group 17-64 years, the percentage of co-infections was the lowest (6.56%, *n* = 8). Mixed infections shown were SARS-CoV-2 + HI (*n* = 2; 1.64%), SARS-CoV-2 + ChP (*n* = 2; 1.64%), SARS-CoV-2 + RSV (*n* = 1; 0.82%), SARS-CoV-2 + PIV3 (*n* = 1; 0.82%), SARS-CoV-2 + HKU-1 (*n* = 1; 0.82%), and triple infection SARS-CoV-2 + HI + RSV (*n* = 1; 0.82%) (Figure 4).

For patients over 65, the percentage of detected mixed infections was 16.35% (*n* = 26). The most frequently proven infectious agent involving SARS-CoV-2 was *Haemophilus influenzae*, found in 13 (8.18%) of clinical samples, while among the viral co-pathogens with the frequency proven was respiratory syncytial virus (RSV) (*n* = 4; 1.89%) (Figure 4). In patients of this age group, triple infections were found: SARS-CoV-2 + *Haemophilus influenzae* + RSV (*n* = 3; 1.26%), SARS-CoV-2 + *Haemophilus influenzae* + PIV2) (*n* = 1; 0.63%), SARS-CoV-2 + AdV (*n* = 2; 1.26%), SARS-CoV-2 + BoV (*n* = 2; 1.26%), SARS-CoV-2 + Chlamydia pneumoniae (*n* = 1; 0.63%), SARS-CoV-2 + HMPV (*n* = 1; 0.63%) and SARS-CoV-2 + NL63 (*n* = 1; 0.63%).

The percentage of co-infections in children under 16 years old was 14.5%, while it was 8.2% in those over 16 years old (*p* = 0.6588). The highest proportion of mixed infections was observed among children aged 6–16 years (33.3%), followed by those under 5 years of age (20.8%). The greatest variety of viral and bacterial co-pathogens was observed in patients over 65 years of age.

### 3.6. Clinical Data of the Investigated Patients

Clinical data from 174/337 (51.6%) patients with COVID-19 were collected from their hospital records. The remaining patients were from emergency departments, and/or clinical data were missing. Patients most often showed symptoms of respiratory disease, such as fever (137 (78%)), fatigue (136 (78%)), cough (131 (75%)), and, less frequently, headache (44 (25%)) and rhinorrhea (59 (34%)). There were 27 (15%) patients with gastrointestinal symptoms, all of whom had diarrhea, and 104 (59.7) of the patients with clinical data had a complication such as pneumonia. The disease ended with a fatal outcome in 9.7% (17/174) of the examined patients, and 2.3% (4/174) were transferred to intensive care.

### 3.7. Age Characteristics of the Clinical Presentation of Respiratory Infection in Mono- and Co-Infected Patients

We examined the disease severity in patients testing positive for both SARS-CoV-2 and another respiratory infection in four age groups: 0–5, 6–16, 17–64, and over 65 (Table 3). Our findings revealed that patients aged 6–16 and over 65 showed the highest incidence of complications such as pneumonia (100% and 88%, respectively), and the mean CRP levels were increased in these two age groups (56% and 78%). Adults over 65 years of age had CRP levels that were approximately five times higher than those in children under 5 years of age (16.2 vs. 78.1; *p* = 0.033). Patients aged 65 and older had the lowest average oxygen saturation levels (below 90%) compared with the other age groups (Table 4). Patients older than 65 years also had a longer hospital stay than did those younger than 5 years (mean: 8.1 ± 3.7 days vs. 3.2 ± 2.2 days; *p* = 0.025).

Blood oxygen saturation levels were decreased in individuals in the 6-16, 17-64, and over-65 age groups co-infected with SARS-CoV-2 compared with those who were mono-infected. The mean SaO_2_% levels were 96% vs. 90.5% (*p* < 0.01), 93% vs. 91.7%, and 90.6 vs. 89.5% for each age group, respectively. We did not observe lower mean levels in co-infected individuals compared with mono-infected individuals among children under 5 years of age (mean SaO_2_ levels: 93.2% vs. 92.7%).

### 3.8. Vaccination Status

The later patients had received only one dose of the vaccine. Vaccination statuses were available for 155 patients. The age range of patients who received at least one dose of the SARS-CoV-2 vaccine was 37 to 92 years (mean age: 69.9 years ± 25.4). Of these patients, 27 (17.4%) were vaccinated. Among those vaccinated, 5 (18.5%) had received one dose, 14 (51.8%) had received two doses, and 8 (29.6%) had received three or more doses. The rate of proven co-infections was higher in vaccinated than unvaccinated individuals (29.6% vs. 15.6%, respectively). The mortality rate was higher in unvaccinated patients than in vaccinated patients (*n* = 2/16 (3.7%) vs. *n* = 14/128 (10.9%); *p* = 0.6926). Among patients with a fatal outcome, the percentage of vaccinated patients with co-infection was higher than that of non-vaccinated patients with co-infection (1/2 50% vs. 1/14 7%; *p* = 0.2417). In patients who did not survive, one vaccinated patient had a mixed infection with *H. influenzae*, and another vaccinated patient had both *H. influenzae* and RSV.

The Cox proportional hazards model (Figure 5) indicated that the risk of death was higher for individuals who were not vaccinated against SARS-CoV-2 and had mixed infections with two or three respiratory pathogens. The risk for unvaccinated patients was 41%, with a *p*-value of 0.713. The risk increased by 2.66 times with co-infection with two pathogens (*p* = 0.342) and by 26 times with three pathogens (*p* = 0.005).

### 3.9. Determining the Clinical Severity of COVID-19 in Mono- and Co-Infections with Bacterial and Viral Co-Pathogens: Treatment

The most frequently reported symptoms characteristic of SARS-CoV-2-positive patients, including fever, cough, fatigue, rhinitis, diarrhea, and headache, were tracked, as shown in Table 4. We found that patients with a positive SARS-CoV-2 test who were co-infected with a co-pathogen virus were more likely to experience headaches than those with a bacterial co-pathogen (44.4% vs. 5.3%; *p* = 0.03). The remaining symptoms showed no significant difference, as shown in Table 5. Regarding laboratory parameters, CRP levels were highest in patients co-infected with three pathogens, with mean levels above 100 mg/L. In addition, these patients also had the most critical blood oxygen saturation levels, with a mean of 87.8%.

Among patients with COVID-19 admitted to ICUs, a higher proportion had co-infection with another respiratory co-pathogen compared with patients with COVID-19 alone (9% vs. 0.7%; *p* = 0.0006). Furthermore, a significantly higher percentage of patients with triple infections was admitted to intensive care than that with double infections (40% vs. 5.3%; *p* = 0.05). The rate of *H. influenzae*-positive patients was significantly higher among those treated in intensive care units (75% (3/4)) compared with those treated in inpatient units (9% (14/153); *p* = 0.004). Furthermore, the rate of patients with triple infections was 50% (2/4) in the intensive care unit compared with 1.3% (2/153) in the inpatient unit (*p* = 0.0029).

The Cox proportional hazards model (Table 5) showed that the risk of death and/or ICU admission was 12 times higher (*p* = 0.042) for those who tested positive for SARS-CoV-2 and were infected with one co-pathogen, which increased by 95% with the appearance of a third (*p* = 0.003). These two factors (occurrence of co-infection) were independently associated with a 29% increased risk of death, with increasing days from the first symptom to hospital admission (*p* = 0.131).

### 3.10. Treatment

Out of 174 patients, 119 (68%) were treated with antibiotics. In a subgroup of 24 patients diagnosed with bacterial infection, 66% received antibiotic treatment. Among the patients with SARS-CoV-2 and co-infection with another viral pathogen, 77% received antibiotic treatment out of a group of nine. Additionally, antiviral drugs were used in 9% of patients (*n* = 15) (Regdanvimab, Remdesivir, or another), corticosteroids in 34% (*n* = 59), vasodilators in 6% (*n* = 11), heparin in 52% (*n* = 92), and oxygen therapy in 44% (*n* = 77).

## 4. Discussion

The COVID-19 pandemic has exposed the vulnerabilities in the health systems of several economically developed countries, including Bulgaria. Hospitals were severely overburdened during the peak spread of the SARS-CoV-2 Alpha and Delta variants [31]. In addition, the possibility of secondary or co-infection with a bacterial pathogen must be considered, which strains healthcare resources. Focusing solely on COVID-19 may cause serious bacterial infections to be overlooked [32]. Therefore, it is essential to identify co-infections in patients with COVID-19. The research shows an association between severe COVID-19, the need for intensive care, and mixed infections involving other respiratory pathogens [25,33,34]. Despite the available literature on such mixed infections, there are gaps in differentiating and comparing the severity of bacterial versus viral co-infections in patients with COVID-19. In addition, data on the clinical presentation of mixed infections with more than two pathogens, especially those involving SARS-CoV-2 and bacterial and other viral respiratory co-pathogens, are lacking or limited. Our study contributes to a better understanding of the clinical severity and etiology of mixed infections with bacterial and viral co-pathogens in SARS-CoV-2-positive patients.

Several studies have found a higher prevalence of bacterial co-infections than viral co-infections in patients testing positive for SARS-CoV-2 [35,36]. In this study, bacterial co-infections were more common than viral co-infection, with rates of 8% and 5.6%, respectively. These findings are consistent with another analysis that showed rates ranging from 3.0% to 9.7% for bacterial co-infections and from 5.4% to 6.6% for viral co-infections [35]. Several studies have shown that *H. influenzae* is the most common bacterial co-pathogen in SARS-CoV-2-positive patients. In our study, it was a common cause of multi-pathogenic co-infection (more than two co-pathogens), with a rate of 80% [37,38,39,40]. The most commonly reported viral co-pathogen with SARS-CoV-2 was RSV, consistent with the findings of this study [41,42]. In continuation of our previous study covering a consecutive period [18], here we also observed the same dependence in the measured Ct value, namely, SARS-CoV-2 had significantly lower values than those of the other respiratory pathogens. Higher Ct values indirectly indicate a lower viral load or relatively attenuated infectivity in respiratory co-pathogens [43]. The results show that SARS-CoV-2 replicates to a greater extent than other respiratory viruses involved in co-infections. Some authors believe that SARS-CoV-2 may dominate over other respiratory viruses in a single-phase infection [44,45]. This may explain why co-infections are less common in patients with SARS-CoV-2. Another study by the team observed an increase in SARS-CoV-2 cases during the winter months of 2021 and spring 2022, with Omicron cases predominating [46].

The curve showing the prevalence of respiratory co-pathogens corresponds to the curve for mono-infections. Variations in this trend are influenced by seasonal and geographic factors and age-related patterns [47,48,49]. Asymptomatic infections with SARS-CoV-2 occur mainly in children [50]. The highest incidence of mixed infections with SARS-CoV-2 and respiratory pathogens was reported in children under 16 years of age (14.2%). as highlighted in a study from 2021 [51].

Co-infections are more common in children under 5 years of age [52], possibly due to the higher incidence of respiratory infections in this age group [53,54]. Several studies have indicated that SARS-CoV-2 infection is less common and less transmissible among children than adults [55,56]. Children who tested positive for SARS-CoV-2 were less likely to require respiratory support and intensive care and had shorter hospital stays than children with other respiratory viruses [57]. Clinical laboratory results show that SARS-CoV-2-positive and co-infected children under 5 years of age usually show mild elevations in CRP levels (mean: 16.2 mg/L), which is common in viral infections [58]. In addition, their average blood oxygen saturation levels are normal or near normal, with a mean of 93.2%. This differs from the usual signs of typical COVID-19 illness [59]. Conversely. these two parameters are typical for COVID-19 in co-infected patients aged over 65 years, showing low saturation (mean SpO2: 89.5%) and high mean CRP levels (mean CRP: 78.1 mg/L). Adults aged ≥65 years remain at increased risk of severe illness from COVID-19 and have higher rates of COVID-19-related hospitalization than other age groups [60]. Additional co-infection can aggravate their condition and lead to a fatal outcome, so prevention of the disease is critical.

The results of this study demonstrate the effectiveness of the vaccine. Despite a higher rate of established co-infections with SARS-CoV-2 in vaccinated patients, their mortality was lower compared with non-vaccinated subjects with mixed infections involving a respiratory co-pathogen. It is important to highlight the likelihood of a high simultaneous prevalence of influenza and SARS-CoV-2 infections in the upcoming winter seasons as the spread curve of both viruses coincides with winter peak times [61]. Developing a combined influenza and SARS-CoV-2 vaccine would be beneficial in preventing deaths and reducing the risk of co-infection. Getting a flu shot can reduce the risk of hospitalization by 29% and the risk of overall death by 18% [62]. Vaccine effectiveness studies have convincingly demonstrated the benefits of COVID-19 vaccines in reducing individual symptomatic and severe illness, which helps lower hospitalizations and ICU admissions [63,64]. The importance of preventing COVID-19 is also highlighted by the significant difference between the percentage of ICU admissions with co-infection and those with mono-infection (9% vs. 0.7%; *p* = 0.0006). Our findings, like those from other studies, indicate that patients with co-infection with COVID-19 are more likely to require intensive care or experience fatality than those with mono-infection [10,34]. This study shows that COVID-19 patients with triple infections experience a more severe form of the disease compared with those with dual infections. The mean CRP levels in co-infections with the three pathogens were above 100 mg/L, and the mean saturation values were critically low at 87.8%. Meanwhile. these clinical indicators are also outside the limits in a mixed infection with two pathogens. but not in such critical limits. Research shows that high CRP levels combined with low saturation indicate a severe form of COVID-19 [59,65,66]. Furthermore, a higher percentage of patients with triple infections require intensive care compared with those with single or double infections. This highlights the importance of using multiplex diagnostics to detect a wider range of pathogens. Detecting co-infections is crucial for determining the cause of the disease and selecting the appropriate treatment [67]. Since symptoms and radiographic images are similar for both bacterial and viral respiratory infections, differentiating between the two can be challenging [68]. This is especially true in cases of co-infections with respiratory pathogens. As a result, the indiscriminate use of antibiotics is concerning, as evidenced by 68% of individuals receiving antibiotics, with only 66% and 77.8% being co-infected with a bacterial and viral co-pathogen, respectively. Implementing multiplex detection of respiratory pathogens in hospitals will likely reduce the emergence of resistant strains.

Despite its contribution to understanding the burden of mixed infections with bacterial and viral co-pathogens, this study had several limitations. The scope of this study was small because patient data were collected from only two hospitals. Clinical data on non-hospitalized patients are lacking, limiting the generalizability of the findings. Future studies should aim to collect data from a wider range of healthcare facilities to broaden the scope of this study. Furthermore, the Ct value is only an indirect method and does not account for the amplification efficiency of the primers for the different respiratory viruses. It should also be noted that nasopharyngeal swabs are not the best approach to look for bacterial infections; sputum culture in patients with pneumonia would be necessary to detect bacterial lower respiratory tract infections.

## 5. Conclusions

In this study, mixed infections with SARS-CoV-2 were found to be relatively rare. Bacterial mixed infections are more common than viral ones. Patients with co-infections have a worse overall clinical condition than do those with a single infection, especially patients over 65 years of age. In addition, the presence of triple infection significantly worsens the condition of patients with COVID-19, leading to a greater need for intensive care and a higher risk of fatal outcomes compared with those infected with two pathogens. It is important to routinely perform multiplex testing of hospitalized patients with COVID-19 to ensure prompt treatment and targeted antibiotic use.

## Figures and Tables

**Figure 1 biomedicines-12-02210-f001:**
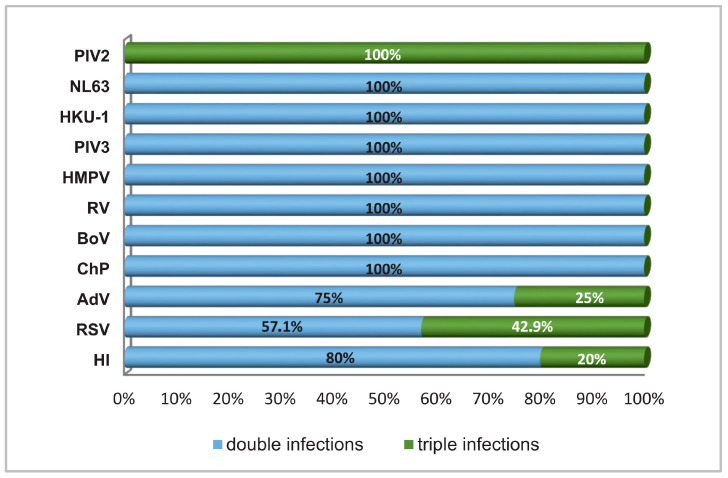
Percentage involvement of individual respiratory pathogens in combinations of double and triple infections: respiratory syncytial virus (RSV), human metapneumovirus (HMPV), parainfluenza virus (PIV) types 2, rhinovirus (RV), adenovirus (AdV), bocavirus (BoV), endemic human coronaviruses (NL63 and HKU1), *Haemophilus influenzae* (HI), and *Chlamydia pneumoniae* (ChP).

**Figure 2 biomedicines-12-02210-f002:**
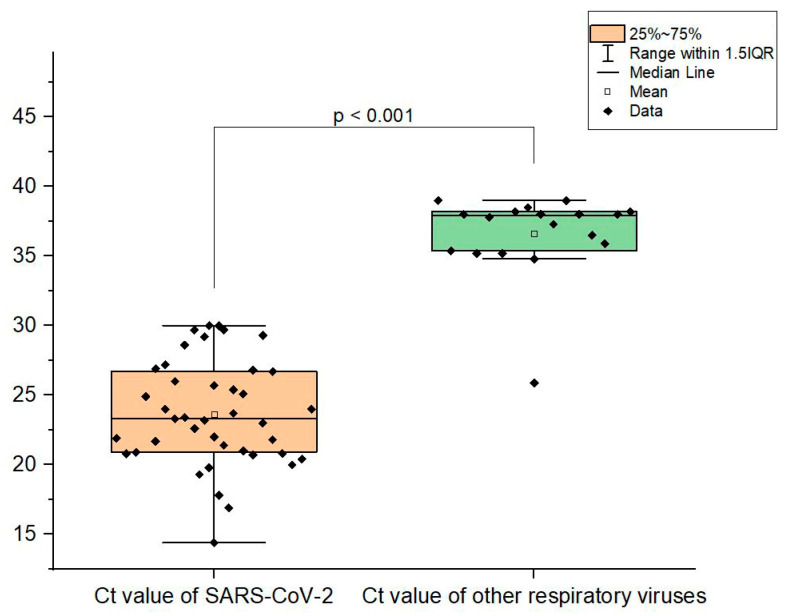
Comparison of the Ct value of viruses in combined infections. Ct was determined for the E gene of severe acute respiratory syndrome coronavirus 2 (SARS-CoV-2). Cycle threshold (Ct) values in the figure are indicated by squares. Mean Ct values, median, and range 25–75% are shown; see Ct distribution. Values were calculated using the Mann–Whitney U-test.

**Figure 3 biomedicines-12-02210-f003:**
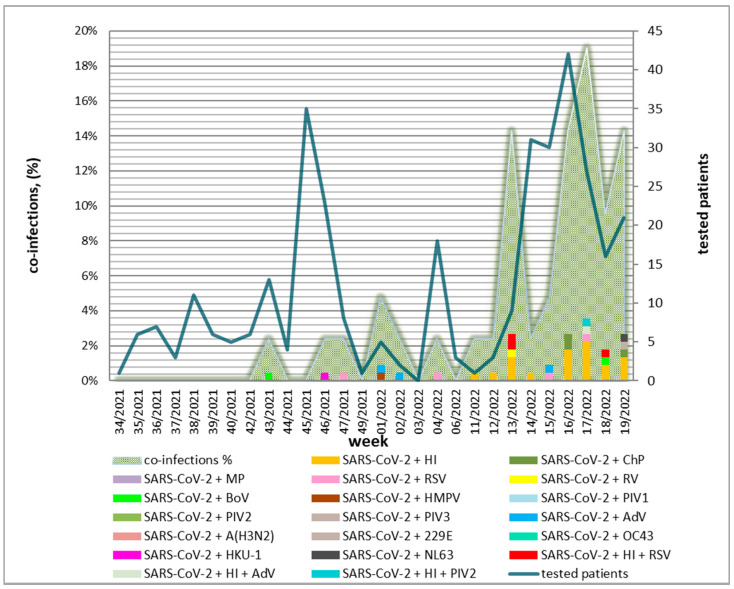
Percentage distribution by week of detected co-infections of the severe acute respiratory syndrome coronavirus 2 SARS-CoV-2 and other respiratory viral and bacterial pathogens (influenza A subtypes A (H3N2), respiratory syncytial virus (RSV), human metapneumovirus (HMPV), parainfluenza virus (PIV) types 1/2/3, rhinovirus (RV), adenovirus (AdV), and bocavirus (BoV), 4 endemic human coronaviruses (HCoVs) (229E, NL63, OC43, and HKU1), *Haemophilus influenzae* (HI), *Mycoplasma pneumoniae* (MyP), and *Chlamydia pneumoniae* (ChP)) in Bulgaria for the period August 2021–May 2022.

**Figure 4 biomedicines-12-02210-f004:**
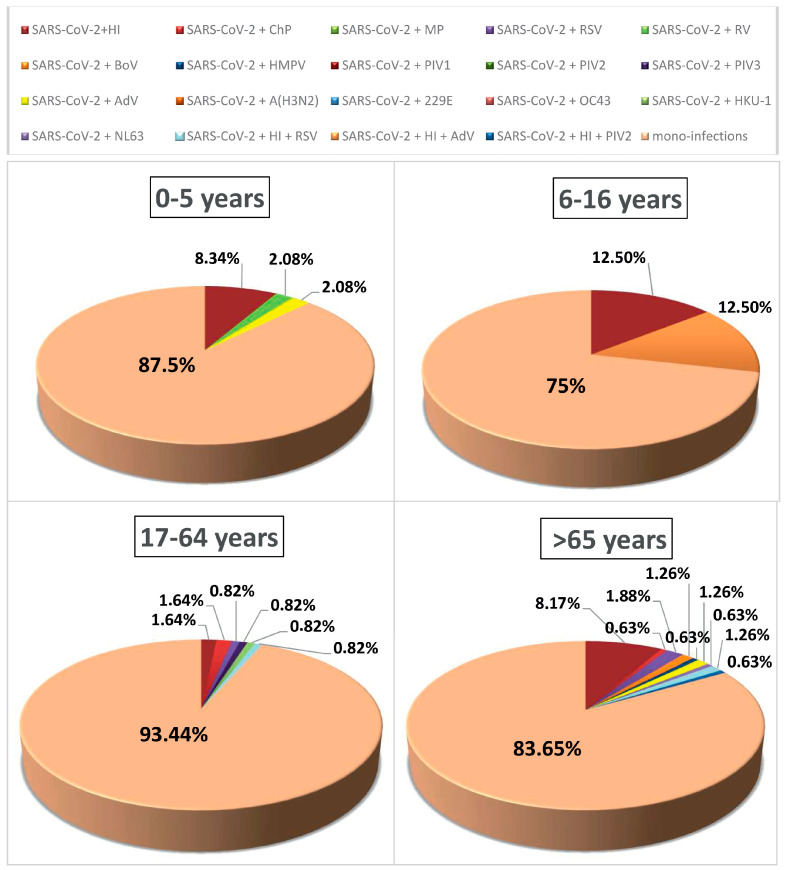
Age distribution of patients co-infected with SARS-CoV-2 and other respiratory pathogens (influenza A subtypes A (H3N2), respiratory syncytial virus (RSV), human metapneumovirus (HMPV), parainfluenza virus (PIV) types 1/2/3, rhinovirus (RV), adenovirus (AdV), and bocavirus (BoV), 4 endemic human coronaviruses (HCoVs) (229E, NL63, OC43, and HKU1), *Haemophilus influenzae* (HI), *Mycoplasma pneumoniae* (MyP), and *Chlamydia pneumoniae* (ChP)). Patients are divided into 4 age groups: 0–5, 6–16, 17–64, and 65 and older. Mono-infected patients were positive for SARS-CoV-2.

**Figure 5 biomedicines-12-02210-f005:**
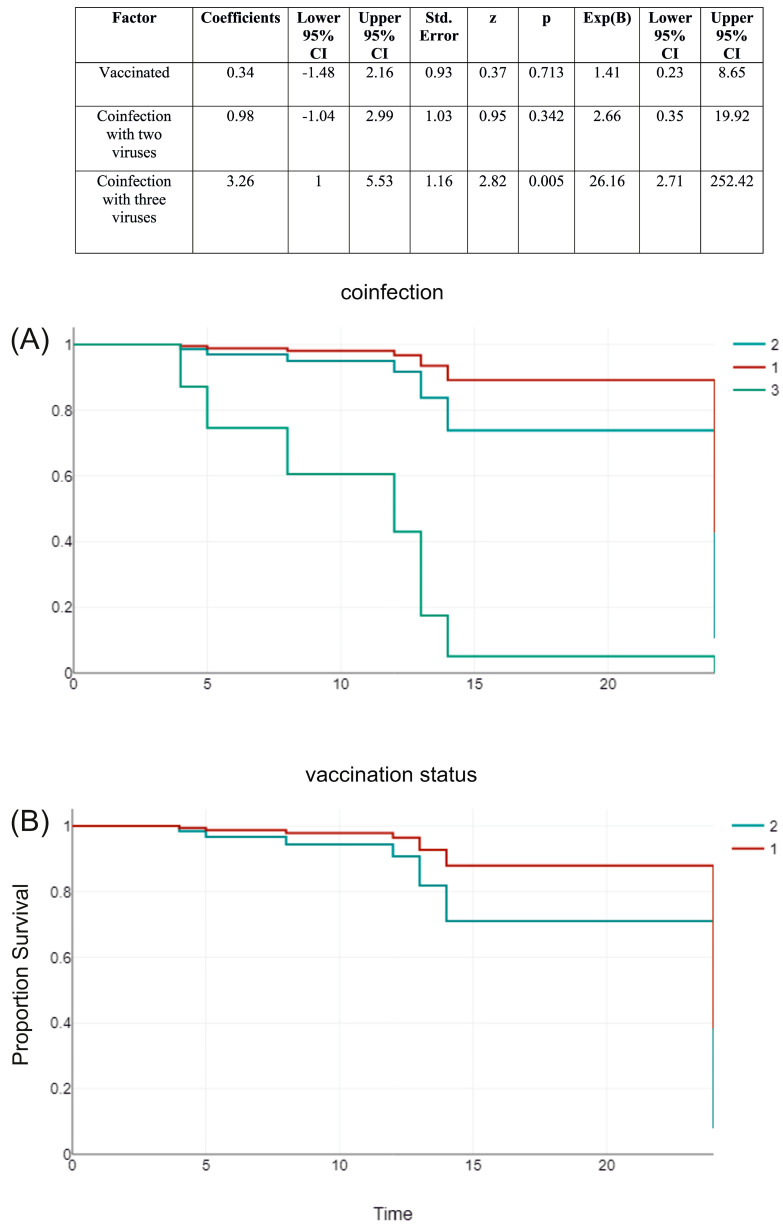
Cox proportional hazards model estimating the risk of mortality in SARS-CoV-2 vaccine recipients compared with unvaccinated subjects, given the presence of co-infection with two or three other respiratory pathogens. Survivor function for groups: (**A**) 1 (red line)—mono-infected, 2 (turquoise line), and 3 (green line); (**B**) 1 (red line)—mono-infected unvaccinated and 2 (turquoise line)—vaccinated.

**Table 1 biomedicines-12-02210-t001:** Proven co-infections with severe acute respiratory syndrome coronavirus 2 (SARS-CoV-2) and influenza A and B; 8 common respiratory viruses: respiratory syncytial virus (RSV), human metapneumovirus (HMPV), parainfluenza virus (PIV) types 1/2/3, rhinovirus (RV), adenovirus (AdV), and bocavirus (BoV), 4 endemic human coronaviruses (HCoVs) (229E, NL63, OC43, and HKU1); and three bacterial pathogens: *Haemophilus influenzae* (HI), *Mycoplasma pneumoniae* (MyP), and *Chlamydia pneumoniae* (ChP).

Co-Infection	*n*	%
SARS-CoV-2 + HI	20	47.62%
SARS-CoV-2 + RSV	4	9.52%
SARS-CoV-2 + ChP	3	7.14%
SARS-CoV-2 + AdV	3	7.14%
SARS-CoV-2 + HI + RSV	3	7.14%
SARS-CoV-2 + BoV	2	4.76%
SARS-CoV-2 + RV	1	2.38%
SARS-CoV-2 + HMPV	1	2.38%
SARS-CoV-2 + PIV3	1	2.38%
SARS-CoV-2 + HKU-1	1	2.38%
SARS-CoV-2 + NL63	1	2.38%
SARS-CoV-2 + HI + AdV	1	2.38%
SARS-CoV-2 + HI + PIV2	1	2.38%
SARS-CoV-2 + MP	0	0%
SARS-CoV-2 + PIV1	0	0%
SARS-CoV-2 + PIV2	0	0%
SARS-CoV-2 + 229E	0	0%
SARS-CoV-2 + OC43	0	0%
SARS-CoV-2 + Influenza A	0	0%
SARS-CoV-2 + Influenza B	0	0%

**Table 2 biomedicines-12-02210-t002:** Distribution according to the severe acute respiratory syndrome coronavirus 2 (SARS-CoV-2) sub-variant among confirmed co-infections with respiratory syncytial virus (RSV), parainfluenza virus (PIV) types 2/3, rhinovirus (RV), adenovirus (AdV), and bocavirus (BoV), endemic human coronaviruses (NL63) *Haemophilus influenzae* (HI), and *Chlamydia pneumoniae* (ChP).

Delta		
Tested: 119		
Co-infections: 2		
Positive rate: 1.7%		
*AY.75.1*	+ BoV	*n* = 1
*AY.4.4*	+ RSV	*n* = 1
**Omicron**		
Tested: 186		
Co-infections: 32 (29 + 3 triple)		
Positive rate: 17.2%		
*BA.1*	+ AdV	*n* = 1
	+ RSV	*n* = 1
*BA.1.1*	+ HI	*n* = 2
*BA.2*	+ HI	*n* = 13
	+ HI + RSV	*n* = 1
	+ HI +PIV2	*n* = 1
	+ HI + AdV	*n* = 1
	+ ChP	*n* = 2
	+ RSV	*n* = 1
	+ BoV	*n* = 1
	+ PIV3	*n* = 1
	+ NL63	*n* = 1
	+ AdV	*n* = 2
*BA.2.12*	+ RV	*n* = 1
*BA.2.9*	+ HI	*n* = 2
	+ ChP	*n* = 1

**Table 3 biomedicines-12-02210-t003:** Distribution of clinical parameters, such as symptoms, laboratory results, length of hospitalization, and clinical outcome, according to age in patients co-infected with SARS-CoV-2 and other respiratory viruses.

Age Group (Years Old)	0–5	6–16	17–64	>65
Co-infected * (*n*)	5	2	7	17
Symptom				
Fever, *n* (%)	4 (80)	1 (50)	7 (100)	13 (76.5)
Fatigue, *n* (%)	4 (80)	2 (100)	7 (100)	11 (64.7)
Cough, *n* (%)	2 (40)	0 (0)	7 (100)	14 (82.4)
Diarrhea, *n* (%)	1 (20)	1 (50)	2 (28.6)	0 (0)
Headache, *n* (%)	0 (0)	0 (0)	0 (0)	3 (17.6)
Rhinitis, *n* (%)	3 (60)	1 (50)	2 (28.6)	4 (23.5)
Pneumonia, *n* (%)	2 (40)	2 (100)	5 (71.4)	15 (88.2)
**Laboratory results**				
Oxygen saturation, mean %	93.2	90.5	91.7	89.5
Lym **,mean × 10^9^/L	2.73	2.1	2.1	2.3
WBC **, mean × 10^9^/L	5.1	9.7	4.9	5.4
CRP **, mean mg/L	16.2	56	42.9	78.1
Hospital stay, mean/SD ** (days)	3.1/2.2	6/5.6	5.2/2.3	8.1/3.7
Hospital stay, median (days)	3	6	6	9
**Clinical outcome**				
ICU ** stay, *n* (%)	1 (20)	0 (0)	1 (14.3)	1 (20)
Fatal outcome, *n* (%)	0	0	1 (14.3)	2 (11.8)

* With clinical data; ** abbreviations used: lymphocytes (Lym); white blood cell (WBC); C-reactive protein (CRP); standard deviation (SD), intensive care units (ICU).

**Table 4 biomedicines-12-02210-t004:** Clinical and laboratory data on patients infected with SARS-CoV-2 and other respiratory viruses and/or bacterial pathogens.

	SARS-CoV-2 Mono-Infection	Viral SARS-CoV-2 Co-Infection	Bacterial SARS-CoV-2 Co-Infection	SARS-CoV-2 Triple Infection	*p*-Value: Viral vs. Bacterial Co-Infection	*p*-Value: Co- vs. Triple Infection	*p*-Value: Mono- vs. Co-Infection	*p*-Value: Mono- vs. Triple Infection
Distribution, *n* (%)	295 (87.5)	14 (4.2)	23 (6.8)	5 (1.5)	–	–	–	–
With clinical data, *n*	141	9	19	5	–	–	–	–
**Symptoms, *n* (%)**								
Fever	110 (78)	8 (88.9)	15 (78.9)	4 (80)	1	1	0.8138	1
Fatigue	111 (78.8)	7 (77.8)	14 (73.7)	4 (80)	1	1	0.8151	1
Cough	107 (75.9)	7 (77.8)	14 (73.7)	3 (60)	1	0.5971	0.8228	0.5971
Diarrhea	22 (15.6)	1 (11.7)	2 (10.5)	2 (40)	1	0.1546	1	0.1895
Headache	39 (27.7)	4 (44.4)	1 (5.3)	–	0.0256	0.5686	0.1825	0.3247
Rhinitis	50 (35.5)	3 (33.3)	4 (21.1)	2 (40)	0.6465	0.5971	0.4198	1
Pneumonia	83 (58.9)	6 (66.7)	12 (63.2)	3 (60)	1	1	0.6955	1
**Laboratory results, mean**								
Oxygen saturation (%)	91.8	93.4	90.4	87.8	n.s *	n.s	n.s	n.s
Lym **, (×10^9^/L)	1.76	0.88	3.1	1.5	n.s	n.s	n.s	n.s
WBC **, (×10^9^/L)	6.9	5.4	5.4	8.7	n.s	0.02	n.s	n.s
CRP **, (mg/L)	65.5	86.7	48.3	109.6	n.s	n.s	n.s	n.s
**Treatment, *n* (%)**								
Antibiotics	96 (68.1)	7 (77.8)	12 (63.2)	4 (80)	0.67	1	1	1
Antiviral drugs	14 (9.9)	–	1 (5.3)	–	1	1	0.3086	1
Corticosteroids	44 (31.2)	3 (33.3)	9 (47.4)	3 (60)	0.687	0.639	0.1525	0.3283
Vasodilators	10 (7.1)	–	1 (5.3)	–	1	1	0.6925	1
Heparin	73 (51.8)	6 (66.7)	11 (57.9)	2 (40)	1	0.6285	0.5658	0.3595
Oxygen therapy	60 (42.6)	3 (33.3)	10 (52.3)	4 (80)	0.4348	0.3353	0.4367	0.1687
**Clinical outcome**								
Hospital stay, mean/SD ** (days)	6.1/4.2	7.8/3.4	5.3/3.3	6.3/2.4	n.s	n.s	n.s	n.s
Hospital stay, median (days)	5	8	4	7				
ICU ** stay, *n* (%)	1 (0.7)	–	1 (5.3)	2 (40)	1	0.0501	0.3129	0.0028
Fatal outcome, *n* (%)	14 (9.9)	–	2 (10.5)	1 (20)	1	0.3996	1	0.4493

* n.s—non-significant; ** abbreviations used: lymphocytes (Lym); white blood cell (WBC); C-reactive protein (CRP); standard deviation (SD), intensive care units (ICU).

**Table 5 biomedicines-12-02210-t005:** The Cox proportional hazards model estimated the risk of mortality and ICU admission in SARS-CoV-2 patients, considering co-infection with two or three pathogens and the associated hazard of increasing days from the first symptom to hospital admission.

Factor	Coefficients	Lower 95% CI	Upper 95% CI	Std. Error	z	*p*	Exp(B)	Lower 95% CI	Upper 95% CI
Co-infection with two viruses	2.5	0.09	4.9	1.23	2.04	0.042	12.14	1.1	134.26
Co-infection with three viruses	4.55	1.51	7.58	1.55	2.94	0.003	94.5	4.54	1967.29
Days from the onset of symptoms before hospitalization	0.25	-0.08	0.58	0.17	1.51	0.131	1.29	0.93	1.79

## Data Availability

This manuscript utilized a database on the distribution of respiratory viruses in Bulgaria, accessible at https://grippe.gateway.bg/index.php (accessed on 26 September 2024).

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
