# Peer review of "Bacterial and Viral Co-Infections in COVID-19 Patients: Etiology and Clinical Impact"

_biomedicines, 2024, doi:10.3390/biomedicines12102210_

Round 1

Reviewer 1 Report

Comments and Suggestions for Authors

1) The authors did not employed RT-PCR, they employed RT-qPCR multiplex. Please correct that part.

2) Bacterial names always goes in italic.

3) Which statistical test was performed in figure 3?

4) About figure 3, please provide a supplementary figure like that but showing the Ct of the endogenous gene. Also performed the same statistical test. Is necessary to evaluated if there was also a statistical significant differences in their endogenous gene Ct-

5) Make a more deep explanation and interpretation of figure 4 and 5. 

6) Table 4 and 6, data about lab parameters please include dispersion data, if es mean add SD, if it is median report percentile 25-75. Also for hospital stay.

7) Figure 6 is lack of statistical analysis. 

Author Response

  • The authors did not emploеd RT-PCR, they employed RT-qPCR multiplex. Please correct that part

.

Thanks, I fixed that

2) Bacterial names always goes in italic.

Thanks, I fixed that

3) Which statistical test was performed in Figure 3?

Comparison of the Ct value of viruses in combined infections. Ct was determined for the ORFab gene of SARS-CoV-2. Ct values ​​are indicated by squares in the figure. Mean Ct values, median, and range 25–75% are shown, see Ct distribution. Values ​​were calculated using the Mann-Whitney U-test

4) About Figure 3, please provide a supplementary figure like that but showing the Ct of the endogenous gene. Also performed the same statistical test. It is necessary to evaluate if there were also statistically significant differences in their endogenous gene Ct-

Кit with which we obtained the Ct values ​​for SARS-CoV-2 e with primers specified for the E gene. This is described in the materials section. We have not used any other method to make a difference.

5) Make a more deep explanation and interpretation of figure 4 and 5. 

Figure 4. Percentage distribution by week of detected co-infections of SARS-CoV-2 and other respiratory viral and bacterial pathogens in Bulgaria for the period August 2021 - May 2022

Figure 5. Age distribution of patients coinfected with SARS-CoV-2 and other respiratory pathogens. Patients are divided into 4 age groups: 0-5, 6-16, 17-64, and 65 and older. Monoinfected patients were positive for SARS-CoV-2.

.6) Table 4 and 6, data about lab parameters please include dispersion data, if es mean add SD, if it is median report percentile 25-75. Also for the hospital stay.

Thanks, I fixed that

7) Figure 6 is lack of statistical analysis.

 The statistical explanation is detailed in the table above graphs A and B. The table in column p explains the statistical significance of 300-304 and the result is explained based on the statistics in the table part of Figure 6.

Reviewer 2 Report

Comments and Suggestions for Authors

Thanks for inviting me to review this manuscript. It is interesting and provides important information. I have the following comments that could be of use:

1.     Line 16 and elsewhere: SARS-CoV-2 and all abbreviations (RSV, ICU, CRP, etc), should be written in full the first time they are used in the abstract, and in the main text

2.     Line 21: ‘’a longer hospital stay (8.1 days; p = 0.025)’’ only one duration of stay is shown here. Please show one vs the other

3.     Line 35: what does ‘aggressive’ mean? Higher mortality? Higher transmissibility? Please explain

4.     Line 40: ‘’Before the 2018–2020 COVID-19 pandemic’’: I don’t understand this. The pandemic started officially in 2020. Please rephrase

5.     Line 54: please change Haemophilus influenza to Haemophilus influenzae

6.     Line 31: as mentioned in comment 1, COVID-19 (and all other abbreviations) should be written in full when first used in the abstract and the main text

7.     Line 88: Please use a reference and explain how you define severe COVID-19

8.     Table S1 and Table S2 are important, but should be removed from the main text

9.     Table 1 and Figure 1: These materials are not important enough to be presented in the main text. They could be moved to the supplementary materials as well

10.  Line 175: This definition is important, but in cases of pneumonia without infiltrates, the appropriate clinical signs and symptoms (such as crackles at the auscultation) may have been missed. This should be mentioned in the limitations section

11.  The institutional review board statement does not mention who approved the study. This should be mentioned, and it should be clear that the approval refers to both hospitals that participated in the study

12.  I am skeptical about why more included patients were hospitalized than those not. Were there any missed cases? This could lead to bias

13.  Line 208, line 244, line 245, line 299, line 326, and maybe elsewhere: H. influenzae should be written in italics

14.  Table 2, Table 3, Table 4, Table 5, Table 6, Figure 2, Figure 4, and Figure 5: A footnote explaining what the abbreviations mean is needed

15.  Figure 3: The Ct values in the other viruses are very high. Such high values make you wonder if they represent true infections

16.  Table 4: What does ‘Rheum’ mean? Please correct this

17.  Line 334-336: This is non-significant and should be erased

18.  Line 341: The authors should explain what antiviral drugs were used. Was it oseltamivir? Was it something else?

19.  Another limitation probably is that the method used to diagnose co-infections is not sensitive enough for all pathogens. A sputum culture in patients with pneumonia would be necessary for lower respiratory tract bacterial infections. Fungal pathogens can also not be detected by the methods used. These should be stated in the limitations subsection of the discussion section

Author Response

 Line 16 and elsewhere: SARS-CoV-2 and all abbreviations (RSV, ICU, CRP, etc), should be written in full the first time they are used in the abstract, and in the main text

Thanks for the note.

Due to the word limits in the abstract, the full names are not described in this section, only the abbreviation. This is in principle permissible in the abstract part to save words

.

  1. Line 21: ‘’a longer hospital stay (8.1 days; p = 0.025)’’ only one duration of stay is shown here. Please show one vs the other

Children <5 years old had more frequent co-infections than adults <65 years old (20.8% vs. 16.4%), while adults showed a more severe clinical picture with a higher CRP level (78.1 vs.16.2 mg/L .; p = 0.033), a lower SpO2 (89.5 vs.93.2%), and a longer hospital stay (8.1 vs. 3.1 days; p = 0.025) (mean levels).

  1. Line 35: what does ‘aggressive’ mean? Higher mortality? Higher transmissibility? Please explain

Pre-Omicron variants of SARS-CoV-2 were relatively more aggressive in terms of more severe clinical presentation of COVID-19 and mortality, and this led to an overload of the hospital system [2].

  1. Line 40: ‘’Before the 2018–2020 COVID-19 pandemic’’: I don’t understand this. The pandemic started officially in 2020. Please rephrase

In 2018–2020, the average prevalence of respiratory viruses reported in studies ranged from 49.8% to 39%, while with the onset of the 2020–2021 COVID-19 pandemic, the prevalence was significantly reduced to 13.4%. [6]

  1. Line 54: please change Haemophilus influenza to Haemophilus influenza

Thanks, I fixed that

  1. Line 31: as mentioned in comment 1, COVID-19 (and all other abbreviations) should be written in full when first used in the abstract and the main text

Thanks I fixed that

  1. Line 88: Please use a reference and explain how you define severe COVID-19

Severe COVID-19 outcomes were defined as defined by the United States Centers for Disease Control and Prevention (CDC) as requiring hospitalization, intensive care unit (ICU) admission, intubation or mechanical ventilation, or death https://www.cdc.gov/covid/hcp/clinical-care/underlying-conditions.html.

  1. Table S1 and Table S2 are important, but should be removed from the main text

Thanks for the note. Thanks, I fixed that.

  1. Table 1 and Figure 1: These materials are not important enough to be presented in the main text. They could be moved to the supplementary materials as well

Thanks I fixed that.

  1. Line 175: This definition is important, but in cases of pneumonia without infiltrates, the appropriate clinical signs and symptoms (such as crackles at the auscultation) may have been missed. This should be mentioned in the limitations section

Thanks. According to the clinicians involved in this study, this observation did not apply to this study.

  1. The institutional review board statement does not mention who approved the study. This should be mentioned, and it should be clear that the approval refers to both hospitals that participated in the study

The statement applies to both hospitals, as they are participants in this project, and this is implied.

  1. I am skeptical about why more included patients were hospitalized than those not. Were there any missed cases? This could lead to bias

Hospitalized patients are more because the aim is to collect more clinical and laboratory data for patients with co-infections. I have also mentioned our other studies involving sentinel surveillance with the theme of co-infections, but they do not have laboratory data, as the GP did not prescribe them.

  1. Line 208, line 244, line 245, line 299, line 326, and maybe elsewhere: H. influenzae should be written in italics

Thanks, I fixed that.

  1. Table 2, Table 3, Table 4, Table 5, Table 6, Figure 2, Figure 4, and Figure 5: A footnote explaining what the abbreviations mean is needed

Abbreviations in tables and diagrams are indicated with their meaning in the text. In each of the tables, abbreviated names of viruses or laboratory indicators are given, written in full in the text.

  1. Figure 3: The Ct values in the other viruses are very high. Such high values make you wonder if they represent true infections

We have a sequencing study of co-infections that proves they are there. Also, the multiplex method we use has a sensitivity of up to 38 cycles, samples are considered positive. It is clear that there are studies that confirm that earlier infection with other non-coronaviruses in our country leads to the appearance of secondary bacterial or viral co-infection. In case of high prevalence of SARS-CoV-2, we excluded co-infection with this virus. There are rare cases of equal reproduction of viruses in general coinfection, and in most cases, one of the two copathogens prevails. I quote our other publications confirming this assertion.

  1. Trifonova I, Christova I, Madzharova I, Angelova S, Voleva S, Yordanova R, Tcherveniakova T, Krumova S, Korsun N. Clinical significance and role of coinfections with respiratory pathogens among individuals with confirmed severe acute respiratory syndrome coronavirus-2 infection. Front Public Health. 2022 Sep 2;10:959319. doi: 10.3389/fpubh.2022.959319
  2. Trifonova, I., Korsun, N., Madzharova, I., Alexiev, I., Ivanov, I., Levterova, V., ... & Christova, I. (2024). Epidemiological and Genetic Characteristics of Respiratory Viral Coinfections with Different Variants of Severe Acute Respiratory Syndrome Coronavirus 2 (SARS-CoV-2). Viruses, 16(6), 958.
  3. Trifonova I, Korsun N, Madzharova I, Velikov P, Alexsiev I, Grigorova L, Voleva S, Yordanova R, Ivanov I, Tcherveniakova T, Christova I. Prevalence and clinical impact of mono- and co-infections with endemic coronaviruses 229E, OC43, NL63, and HKU-1 during the COVID-19 pandemic. Heliyon. 2024 Apr 7;10(7):e29258. doi: 10.1016/j.heliyon.2024.e29258. PMID: 38623185; PMCID: PMC11016702.
  4. Table 4: What does ‘Rheum’ mean? Please correct this

Thanks, I fixed that.

  1. Line 334-336: This is non-significant and should be erased

I have corrected the sentence, although the result is statistically insignificant, it is important for readers to clarify the interpretation of gamble regression as a statistical method.

These two factors (occurrence of co-infection) were independently associated with a 29% increased risk of death, with increasing days from the first symptom to hospital admission (p = 0.131).

  1. Line 341: The authors should explain what antiviral drugs were used. Was it oseltamivir? Was it something else?

Additionally, antiviral drugs were used in 9% of patients (n = 15)(Regdanvimab , Remdesivir or another ), corticosteroids in 34% (n = 59), vasodilators in 6% (n = 11), heparin in 52% (n = 92), and oxygen therapy in 44% (n = 77).

  1. Another limitation probably is that the method used to diagnose co-infections is not sensitive enough for all pathogens. A sputum culture in patients with pneumonia would be necessary for lower respiratory tract bacterial infections. Fungal pathogens can also not be detected by the methods used. These should be stated in the limitations subsection of the discussion section

I agree that nasopharyngeal swabs are not the best sampling selection for bacterial infections, but in this study, we were not investigating whether fungal infections cause respiratory disease.

    Despite its contribution to understanding the burden of mixed infections with bacterial and viral co-pathogens, this study had several limitations. The scope of this study was small because patient data were collected from only two hospitals. Clinical data on non-hospitalized patients are lacking, limiting the generalizability of the findings. Future studies should aim to collect data from a wider range of healthcare facilities to broaden the scope of this study. Furthermore, the Ct value is only an indirect method and does not account for the amplification efficiency of the primers for the different respiratory viruses. It should also be noted that nasopharyngeal swabs are not the best approach to look for bacterial infections, sputum culture in patients with pneumonia would be necessary to detect bacterial lower respiratory tract infections.

Round 2

Reviewer 1 Report

Comments and Suggestions for Authors

The authors uses without distinction coma (,) or period (.) to separate decimales. For example Table 5 and 6. Please, use . to separate decimales. 

Previously I asked a deeper explanation and interpretation of figure 4 and 5, but this was no make. Please make a deeper o better interpretation of the dose results. 

I can not found figure 6, now is table 5?

Author Response

The authors uses without distinction coma (,) or period (.) to separate decimales. For example Table 5 and 6. Please, use . to separate decimales. 

Thanks, I fixed that

  1. Previously I asked a deeper explanation and interpretation of figure 4 and 5, but this was no make. Please make a deeper o better interpretation of the dose results.

Naw fig 4 is 3 and 5 is 4:

Fig. 3

The study period spanned a total of 28 weeks (August 2021–May 2022) (Figure 3), with a greater percentage of study samples collected during weeks 13–19 of 2022. The highest rates of co-infections were found during the period 13 -19 weeks of 2022. During the different periods of the study, different combinations of other participants in mixed infections were observed. During weeks 34 to 42 of 2021, no mixed infections were identified. A high rate of co-infected patients with SARS-CoV-2 was demonstrated during weeks 13, 16, 18, and 19 of 2022 (March-May). The highest number of co-infections were detected in week 17 of 2022 (April 25-May 1). The same week also saw the greatest diversity in terms of proven co-infections: SARS-CoV-2 + Haemophilus influenzae (n=5), SARS-CoV-2 + RSV (n=1), SARS-CoV-2 + Haemophilus influenzae + AdV (n=1) and SARS-CoV-2 + Haemophilus influenzae + PIV2 (n=1).

Fig.4

four age groups to analyze their age distribution: 0-5, 6-16, 17-64, and 65 years and older ( Figure 4). n the group of the youngest children (0-5 years), positive for SARS-CoV-2, 6 (12.5%) cases of co-infections involving Haemophilus influenzae were proven (n=4; 8.33 ) %), rhinovirus (RV) (n=1; 2.8%) and adenovirus (AdV) (n=1; 2.8%) (Figure 10-a). In children and adolescents aged 6-16 years, the percentage of mixed infection is the highest (25%), with the following combinations established: SARS-CoV-2 + Haemophilus influenzae (n=1) and SARS-CoV-2 Haemophilus influenzae + AdV (n=1) (Figure 4).

In the age group 17-64 years, the percentage of co-infections was the lowest (6.56%, n=8). Mixed infections shown are respectively: SARS-CoV-2 + HI (n=2; 1.64%), SARS-CoV-2 + ChP (n=2; 1.64%), SARS-CoV-2 + RSV ( n=1; 0.82%), SARS-CoV-2 + PIV3 (n=1; 0.82%), SARS-CoV-2 + HKU-1 (n=1; 0.82%), and triple infection SARS-CoV-2 + HI + RSV (n=1; 0.82%) (Figure 4).

 For patients over 65, the percentage of detected mixed infections was 16.35% (n=26). The most frequently proven infectious agent involving SARS-CoV-2 was Haemophilus influenzae, found in 13 (8.18%) of clinical samples, while among the viral co-pathogens with the highest frequency proven respiratory syncytial virus (RSV ) ) (n=4; 1.89%) (Figure4). In patients of this age group, triple infections were found: SARS-CoV-2 + Haemophilus influenzae + RSV (n=3; 1.26%), SARS-CoV-2 + Haemophilus influenzae + PIV2) (n=1; 0.63%), SARS-CoV-2 + AdV (n=2; 1.26%), SARS-CoV-2 + BoV (n=2; 1.26%), SARS-CoV-2 + Chlamydia pneumoniae (n=1; 0.63%), SARS-CoV-2 + HMPV (n=1; 0.63%) and SARS-CoV-2 + NL63 (n=1; 0.63%).

3.I can not found Figure 6, now is table 5?

figure 6 naw is Figure 5. :Cox proportional hazards model estimating the risk of mortality in SARS-CoV-2 vaccine recipients compared with unvaccinated subjects, given the presence of co-infection with two or three other respiratory pathogens. Survivor function for groups: A) 1 (red line)—mono-infected, 2 (turquoise line), and 3 (green line); B) 1 (red line)—mono-infected unvaccinated and 2 (turquoise line)—vaccinated.

Reviewer 2 Report

Comments and Suggestions for Authors

I insist on comments 1 and 14, as shown in the previous review report.

Comments on the Quality of English Language

Minor

Author Response

I insist on comments 1 and 14, as shown in the previous review report.

Thanks for the review, I agree with the comments.
1. All abbreviations in the abstract are explained.
14. All abbreviations in tables and diagrams are mentioned either in the title of the figure/table or in the accompanying notes.
